# Students with Autism Spectrum Disorders and Their First-Year College Experiences

**DOI:** 10.3390/ijerph182211822

**Published:** 2021-11-11

**Authors:** Stefania D. Petcu, Dalun Zhang, Yi-Fan Li

**Affiliations:** 1Department of Individual, Family & Community Education, College of Education & Human Sciences, University of New Mexico, Albuquerque, NM 87131-0001, USA; 2Center on Disability and Development, Department of Educational Psychology, Texas A&M University, College Station, TX 77843-4225, USA; dalun@tamu.edu (D.Z.); wagala23@exchange.tamu.edu (Y.-F.L.)

**Keywords:** autism, postsecondary education, learning disabilities

## Abstract

Using data from the 2019 CIRP Freshman Survey and the Your First College Year (YFCY) from the Higher Education Research Institute at UCLA, this study explores the differences between the characteristics and behaviors of the first-year students with autism spectrum disorders (17) and those of students with learning disabilities (102). The findings indicate that the characteristics of these two groups of first-year college students were similar except for gender, ethnicity, first college generation, and parents’ income. Compared with first-year college students with LD, students with ASD were less likely to engage in risk-taking behaviors, use health services and the writing center.

## 1. Students with Autism Spectrum Disorders and Their First Year College Experiences

The prevalence of autism spectrum disorders (ASDs) has steadily increased over the past few decades [1]. One in 54 American school-aged children is diagnosed with ASD [2]. ASD is a developmental disability that usually begins before 3 years of age, and it is characterized by persistent deficits in three distinct areas of social communication and interactions in addition to the occurrence of at least two of four types of repetitive behaviors [3]. Youth with ASDs must be provided with services and support that promote positive post-high-school outcomes, including employment and postsecondary education. Meaningful progress has been made during the past four decades to facilitate access to higher education for students with disabilities [4]. Compared with their peers with other disabilities, students with ASD are less likely to take steps to prepare for college while in high school [5] and more likely to enroll in a 2-year college (32%) than a 4-year college (17%). Moreover, the postsecondary completion rates for students with ASD remain significantly lower (39%) than that of their peers in the general population (59%) or of that for students with all types of disabilities (50%) [6]. This low completion rate is similar to findings of previous research showing that students with ASD in higher education struggle more than their peers with other disabilities or without disabilities, mainly dealing with higher rates of loneliness, anxiety, depression, and an increased rate of dropping out before degree completion [7].

The low college participation and completion rates result in high economic and personal costs for youth with ASDs, their families, and society. Based on national high-school graduation estimates [8], the prevalence rate suggests that approximately 49,000 youths with ASDs will have graduated from high school from 2014 to 2015. Yet, almost 33,300 of them would not pursue any postsecondary education after leaving high school in the first several years. Some may not pursue a college but only take a vocational course of study [6]. The U.S. Bureau of Labor Statistics reports that workers with a bachelor’s degree earn an average of USD 381 per week more than workers with some college experience but with no degree or no college education, a difference of almost USD 600,000 over a 30-year working life. Thus, it is critical to encourage and support students with ASD to pursue a college education. We need to better understand the current college students with ASD and how they perform in college, especially during their first year of postsecondary education.

Research on college persistence and retention show that the success of students without disabilities is affected by a combination of individual and family characteristics coupled with higher education institution characteristics [9]. Three categories of factors act as facilitating factors or barriers to the retention of college students. These include situational factors such as physical health, employment, and family responsibilities, institutional factors such as policy, support programs, financial aid, and staff responsiveness, and dispositional factors such as self-confidence, attitudes, and beliefs [10,11]. Early identification of students at risk of dropping out during the first year of college can lead to early intervention actions such as tutoring, counseling, and mentoring, all of which prevent student withdrawal [12,13,14,15,16,17].

For students with disabilities, methods for increasing college retention and success have focused on providing access to both physical and other disability-related resources (e.g., classroom accommodations). However, access to disability-related resources is conditioned by student’s self-disclosure of their disability. Students with disabilities transitioning to higher education encounter a new legal framework under which the responsibility for accessing disability-related supports shifts from the school to the student. In high school, special education is governed by the Individuals with Disabilities Education Act of 2004 [18]; schools are mandated to identify and provide supports and accommodations students may need. In higher education, students with disabilities are covered by two civil rights laws—Subpart E of Section 504 of the Rehabilitation Act of 1973 [19], and the Americans with Disabilities Act of 1990 [20] and its Amendments Act of 2008 [21]. However, disability disclosure in higher education is voluntary, and only 24% of students with ASD notify their school of their disability [22]. Without disclosure, students are not eligible for disability-related accommodations, despite developing research showing that accessing accommodations can make a positive difference in the academic experience of students with disabilities, including those with ASD [23,24].

There are various reasons why students do not disclose their disabilities, including difficulties in understanding and navigating the process or a conscious choice made by the student [25]. Additional potential reasons include lack of awareness of the availability of accommodations [26], which can potentially be the result of insufficient disability-related training for faculty and instructional staff [26,27,28]. Some students with ASD delay disclosure of their disability, which also triggers a delay in appropriate support. Such delay leaves more room for challenges and increases the possibility of dropping out [25,29]. However, college students with disabilities who received support—that which is available to the entire student body and disability-specific support—were more likely to persist in and complete a 2-year or 4-year college course [22]. Thus, the low rate of disability disclosure and underutilization of accommodations and support services may contribute to a lower degree-completion rate [30].

Some common characteristics of students with ASD present additional challenges in their transition to college. For example, social impairments, communication difficulties, and repetitive behaviors [3] may lead to academic, social, and daily living challenges as they transition to college. These core challenges and the comorbid conditions that often accompany ASD might reduce the likelihood that young adults can successfully attain the social roles that mark a normative transition from adolescence to adulthood. Some of these comorbid conditions include mental health conditions such as anxiety [24,29,31,32] and depression [33]. These individuals might also encounter hypersensitivity to light [34,35] or poor executive functioning [36]. All of these together create unique challenges that students with ASD must face in their transition to college life in addition to the traditional challenges that all students face [37]. Moreover, college students with ASD have indicated some areas of concern about their college life such as coursework requirements, disability awareness, availability of practical and emotional support, organizational difficulties such as time-management and problem-solving skills, sensory challenges, and mental health [38,39,40,41,42]. Students with ASD reported poorer self-rated physical and mental health, more depressive symptoms, and more signs of anxiety than other students [43].

Beyond the educational challenges of entry requirements and the shift to a new legal framework, students need to develop a level of independence during the first year of college. They need to learn the necessary skills for managing accommodations, finances, catering, and meeting increased demands [44]. Students with ASD could benefit from social supports, guidance, and explicit instructions to follow during the counseling process [28]. Some universities and colleges offer a network to support students with disabilities, including ASD. Boney, Potvin, and Chabot [45] developed a collaboration program between the Office of Student Accessibility and the occupational therapy department. The program employed coaching to help students achieve their self-identified goals, such as time management, health and wellness, and academic-related goals. Evaluation of the program showed that 80% of students with disabilities achieved their goals. Students also expressed that the program was supportive of their college learning. Weiss and Rohland [46] introduced the Communication Coaching Program (CCP) to support students with ASD. The CCP program utilized a diverse support system, including disability counseling, communication coaching, peer coaching, social groups, and campus resources. Students in this program showed improvement in executive functioning and social communication skills.

Peer mentoring is another effective strategy to support students with ASD in higher education. Siew et al. [47] implemented a peer-mentoring program for students with ASD. The peers in the program were considered as “specialist peer mentors” because they had experienced and professional backgrounds in working with individuals with ASD. The pre-and post-test results showed that students with ASD in this program demonstrated improvement in several areas, including adjustment to the university and communication skills. Similarly, Ncube et al. [48] introduced the Autism Mentorship Program (AMP) to support students with ASD to achieve their goals. A total of 23 students with ASD participated in the AMP program during their first year. Students expressed that the AMP program supported their goal development and achievement. This study also highlighted the importance of providing support to students with ASD in their first-year college experience.

Student success and retention continue to be of concern for higher education institutions. Broader participation, combined with lower completion rates for nontraditional students, highlights the need for new ways of understanding the student experience to ground policy and practice. The first year of college plays an essential role in students’ adjustment to college life, but it is also a great indicator of student retention and success [49,50]. The CIRP Freshman Survey and the Your First College Year (YFCY) provide a unique opportunity to explore the characteristics of college students and their behaviors during their freshman year. The CIRP Freshman Survey is designed to be administered to incoming first-year students before they start their first college semester, and the main purpose of the survey is to collect information that will allow higher education institutions to know who their incoming students are before their college experience. Your First College Year (YFCY), it is a follow-up of the CIRP Freshman Survey, and it is specially designed to measure student development in the first year of college, and it is administered at the end of the first year. Data from the 2019 CIRP Freshman Survey & YFCY indicate that 68% of students rated their academic ability above average; 79% of them sought academic advice and 26% of them sought personal counseling [31]. Higher education institutions use such types of data to understand who their incoming students are and how they behave in their first year of college in order to better design the support they need to return for their second year and to successfully graduate [47].

Therefore, it is essential to explore how students with ASD behave during the first year of their higher education program. The CIRP Freshman Survey and the Your First College Year (YFCY) provide a unique opportunity to explore the characteristics of students with ASD and how they behave during the critical first year in college. Moreover, data from these two surveys allow us to explore the differences between college students with ASD and those with Learning Disabilities. Preliminary research on college students with ASD pinpoints some specific challenges, and thus, specific needs to help them be successful in college [38,39,40,41,42,44]. Additionally, the law regarding the provision of services for students with disabilities in secondary education [40] clearly requires that services must be tailored to the specific needs of the students. However, there is a gap in the literature in understanding who the college students with ASD are, and also how they are different from their peers with other disabilities. Such differences are essential in determining the need for more individualized and adapted college support for students with ASD [51,52]. Postsecondary education students with ASD are more likely to disclose their disability with their office of disability than their peers with LD. The odds that students with ASD received postsecondary accommodations and other disability-specific supports were almost six times those for students with LD [41]. However, the number of students with ASD who enroll and graduate from different types of postsecondary education programs is lower than that of students with LD [41,53]. During high school, students with ASD are more likely to have their parents attend their Individualized Education Plan (IEP) meeting than their peers with LD (93% vs. 83%). However, students with ASD report being slightly less likely to have their interests and strengths discussed during the IEP meeting than their peers with LD (91% vs. 95%) or to receive information on education, career, and living options for after high school (54% vs. 64%) [53]. The postsecondary education expectations of students with ASD, or their parents, are lower than those of students with LD (75% vs. 86%; 53% vs. 78%) [53]. Given the importance of the first year in college for retention and success [49,50] and the differences in college success between students with ASD and those with LD [41,53], it is essential to understand how first-year college students with ASD are different to those with LD and how their behaviors differ. Hopefully, such understanding will allow for the provision of supports better tailored to the needs of first-year college students with ASD. Hence, our study aimed to answer the following research questions:

What are the individual and familial characteristics of college students with autism spectrum disorders, and how are these different from those of other college students with learning disabilities?

What type of supports are accessed by college students with autism spectrum disorders, and how are these different from those accessed by other college students with learning disabilities?

## 2. Materials and Methods

Data for this study were retrieved from the 2019 CIRP Freshman Survey and the Your First College Year (YFCY). Both surveys collect data on students with disabilities in odd years. The CIRP Freshman Survey is designed for administration to incoming first-year students before they start college. In 2019, the survey included 95,505 first-time, full-time (FTFT) first-year students entering 148 baccalaureate institutions across the United States who responded to this survey. The survey collects extensive information that allows higher education institutions to know their students before they experience college. Critical sections of this survey examine the following: established behaviors in high school, academic preparedness, admissions decisions, expectations of college, interactions with peers and faculty, student values and goals, student demographic characteristics, and concerns about financing college. Many of the CIRP Freshman Survey items are pretest questions included in the CIRP follow-up surveys titled Your First College Year (YFCY), which provides a longitudinal examination of cognitive and affective growth during college. The CIRP Freshman Survey is the only survey that includes a question asking participants to disclose their disability; because the same participants receive the follow-up survey YECY at the end of the first year, these two surveys provided longitudinal data on how students with ASD behave during their first year in college.

The YFCY was developed through a collaboration between the Higher Education Research Institute at UCLA and the Policy Center on the First Year of College at Brevard College to identify features of the first year that promote students’ learning, involvement, satisfaction, retention, and success. Any colleges wishing to improve their first-year programs and retention strategies can use these data. Specifically, the survey collects information on several academic experiences of first-year students, including classroom activities, academic engagement and disengagement, and interaction with faculty. Additionally, several questions on YFCY assess students’ experiences with various campus programs such as orientation, honors courses, first-year seminars, remedial coursework, service-learning opportunities, academic advising, learning communities, and interaction with campus advisors, counselors, and other support personnel. YFCY is designed to help institutions assess how their students have changed since entering college as a follow-up instrument. When combined with CIRP Freshman Survey data, the YFCY serves as a longitudinal measure of students’ cognitive and affective growth during the first year. More information about YFCY and CIRP Freshman Survey can be found at this address CIRP Freshman Survey https://heri.ucla.edu/instruments/ (accessed on 8 November 2021).

### Data Analysis

We used Cross Tabulation tests to examine whether there was a relationship between two categorical variables. One of the benefits of using Cross Tabulation was that it demonstrated a breakdown of the data by using two categorical variables, and chi-squared tests showed whether the relationship of the two categorical variables were statistically significant or not. In this study, we intended to examine the relationship between disability status and the following variables: (a) first-year interactions (academic advisors/counselors, close friends at this institution, your siblings or extended family, close friends not at this institution, faculty during office hours, faculty outside of class or office hours, graduate students/teaching assistants, your parents/guardians). (b) Self-rating (risk-taking, self-confidence (intellectual), self-confidence (social), and understanding of others). (c) act in college (taken a remedial or developmental course, participated in an academic support program, discussed course content with students outside of class, received tutoring, worked with classmates on group projects, working (for pay) on campus and working (for pay) off campus). (d) Services (study-skills advising, financial aid advising, student health services, student psychological service, writing center, disability resource center, career services, and academic advising).

The Fisher exact test was used to assess the association between disability status and 1st-year college behaviors and accessed support services. The Fisher exact test is an appropriate test for small samples because it is calculated using the exact null randomization distribution, which, for small samples, can substantially differ from the distributions assumed by other parametric tests [54]. Because of small sizes, we reduced the number of categories for the outcomes to two to maintain the power to detect differences.

## 3. Results

### 3.1. The Individual and Familial Characteristics of College Students with Autism Spectrum Disorders and Their Difference from Those of Other College Students with Learning Disabilities

Participant demographic characteristics are shown in Table 1. Of all the students who reported having ASD, 65% are males, and 26% are females; the majority are Whites. Among them, 74% were enrolled in a four-year university program, with 50% expecting a master’s degree to be their highest level of education. Most of them were full-time students, and 17% reported being first-generation college students. Overall, the characteristics of students with ASD are similar to those with LD, with few exceptions (e.g., gender, ethnicity, the expected level of education, or being the first generation to go to college in the family) even though the LD sample was larger than that of students with ASD.

As part of the 2019 YFCY surrey, students were asked to rate themselves on several traits in comparison with the average person their age. Respondents were asked to select only one of the following options: highest 10%, above average, average, below average, and lowest 10%. Self-confidence (intellectual) was the only self-rating category for which students with ASD reported a higher level than students with LD (See Table 2). Students with ASD reported low scores on risk taking. The *p*-value (0.04) of the Fisher exact tests indicates that students with ASD are significantly different from those with LD, showing that students with LD are more likely to exhibit risk-taking behaviors during their first year of college than their peers with ASD.

A large majority of first-year students with ASD and LD reported not working for pay on or off campus; only a small percentage of students with LD reported working for pay on campus. Table 3 shows that the majority of both students with ASD and LD reported not taking a remedial or developmental course. More than half of the students with LD and 27% of those with ASD reported being enrolled in an academic support program during their first year of college. However, the *p*-value of the Fisher exact test was not significant.

### 3.2. Types of Support Accessed by College Students with Autism Spectrum Disorders and Their Difference from Those Accessed by Students with Other Types of Disabilities

Table 4 shows the kind of people students with ASD and those with LD interact with during their first year in college. Students with ASD reported interacting most often with close friends at their institution, siblings, and immediate family, close friends not at this institution, and their parents. This pattern of interactions was similar to that of students with LD. Students with ASD reported fewer interactions with faculty, academic advisors, and graduate students during their first year. While the patterns of the academic interactions during the first year were similar for the two groups, students with ASD seemed to interact more often with graduate students than with LD. No Fisher test was significant.

Students with ASD and those with LD can use various support services available at their higher education institution during their first year (Table 5). A vast majority of the students with ASD reported not using financial aid services, psychological services, or the writing center. While students with LD had similar use of financial aid and psychological services, a larger percentage of them reported using the writing center. About a quarter of the students with ASD reported frequently using the disability center and being involved in group projects. Students with LD, on the other hand, reported discussing courses outside the class and being involved in group projects more often. There were no significant differences between students with ASD and those with LD regarding their use of support services, except for health services (*p*-value of 0.01) and the writing center (*p*-value of 0.04). These values indicate that students with LD are more likely to use both health services and the writing center during the first year of college.

## 4. Discussion

This study had two aims: to identify the characteristics of first-year college students with autism spectrum disorders and the type of support services they are using; and an additional aim was to examine the differences in terms of characteristics and behaviors between them and their peers with learning disabilities. The unique characteristics of students with disabilities add to the challenges of their first year in college compared to students with other types of disabilities or those without disabilities. As the number of college students with disabilities increases, their graduation rates continue to be low compared to peers without disabilities [5]. Thus, is critical to explore how to better address the needs of college students with ASD, particularly during the first year of college when they need to learn how to adjust to a new culture where they need to identify the resources they need to successfully respond to the high demands of college life.

Overall, the findings of this study indicate that the characteristics of students with ASD and those with LD were similar with few exceptions (gender, ethnicity, first college generation, and parents’ income). These findings support previous findings that showed that the characteristics of college students with ASD were similar to those with other disabilities [51]. On the contrary, Elias and White [52] found that the challenges of college students with ASD were distinct compared with those of students with ADHD.

The differences in the demographic characteristics between the two groups included in this study are worthy of further exploration. For example, 65% of students with ASD in this sample were males, while the dominant gender among students with LD was female. Moreover, 85% of students with LD were White, with a very small percentage of students representing other races/ethnicities. Students with ASD, on the other hand, included more students from different races/ethnicities, although the majority were still White. Previous research indicated that demographic characteristics should be examined to find disparities between students with ASD and students with other disabilities. For example, Anderson et al. [55] demonstrated that students with ASD presented different living arrangements compared to students with other disabilities. They indicated that demographic characteristics of individuals with ASD helped explore what was unique in the population.

Similarly, Anderson et al. [55] conducted a systematic literature review on postsecondary students with ASD interventions. The study examined the characteristics of study participants based on 24 empirical studies. Findings from this study suggested that efforts to address the diverse needs of students with ASD must consider students’ characteristics, including the severity of ASD symptoms and other comorbidity issues. Thus, it is critical that the field continues to explore the characteristics of postsecondary students with ASD and continues to consider cultural identities while providing in-college support, especially in the first year.

Given the characteristics of ASD, it is not surprising that students included in this study reported low levels of risk-taking or social aspects of self-confidence. These findings might support previous studies that found postsecondary students with ASD are prone to loneliness, anxiety, and depression [56,57]. However, this study found that students with ASD report higher rates of the intellectual aspects of self-confidence. This supports the findings of other studies [58] that report postsecondary students with ASD enrolling highly in STEAM majors. Compared to students with ASD in non-STEAM fields, students with ASD in STEAM fields were more likely to persist and succeed in college [58]. We further explained that ASD’s innate characteristics (e.g., the preference for rule-based and structured systems) might be why students with ASD in STEAM fields tended to complete college or transfer from 2-year community colleges to 4-year universities.

Social interactions were found to play an important role in college retention and success. This study found that students with ASD largely interacted with friends and family but less with faculty. Barnhill [59] also identified one of the challenges for students with ASD was that faculty might not recognize the characteristics of ASD. Therefore, Barnhill suggested that providing concise and valuable information regarding ASD for the faculty is necessary. Further research needs to explore the extent to which interactions with faculty are associated with retention and graduation for students with disabilities.

Previous studies have argued that the support offered by higher education disability offices might not address the needs of students with ASD and call for institutions to identify and provide supports beyond those offered by the office of students with disabilities [53]. Both students with ASD and LD included in this study reported not using the disability offices’ supports very much. This finding is not surprising given that only about a quarter of students with disabilities disclose their disability. Because students with ASD need to access both disability-related supports and supports for all students [60], higher education institutions should expand their efforts to address student diversity and find more effective ways to reach out to all students to help them identify and use all available resources to help improve retention and graduation. In addition to access disability-related supports and supports, Barnhill [59] reminded researchers and relevant stakeholders to explore whether the services or supports provided for students with ASD are helpful or not. Barnhill used a survey to investigate on-campus supports for students with ASD. Although support for social skills training was valuable, the study showed that social skills instructions could be challenging and ineffective. For example, students with ASD complained that the teaching of social skills was repetitive. Students with ASD might receive similar instruction in high school. The study indicated that future research needs to explore the methods that effectively design services and supports for students with ASD, and address diversity needs from this population.

## 5. Limitations

There are several significant limitations of the present study that should be acknowledged. First, the low number of students with ASD included in this study means that its power is relatively low regarding the detection of differences, and some that are not significant might still be meaningful. To address this limitation, we used the Fisher exact test to examine the significance of the differences between the two groups in their use of support services. Despite a low number of students with ASD included in this study, this exploratory study raises awareness about the importance of better understanding who students with ASD are as soon as they start their first year in college. Another limitation is that we performed multiple hypothesis tests to identify differences between students with ASD and those with LD, and we did not correct for multiple tests. If a correction for multiple tests had been used, some of the significant differences seen would not have been significant. The reason for not using corrections was that power was already low, and we did not want to miss meaningful differences; however, the significant differences are somewhat less likely to be replicated in future research. Lastly, the data used in this study are self-reported; thus, the responses to survey items could not be independently verified. Despite the limitations mentioned above, the findings of this study confirm the need to continue exploring the characteristics of college students with ASD and those from different disability categories. Notably, further research should explore how students with ASD behave in their first year of college and how better guidance should be provided to higher education institutions and their disability support personnel.

## 6. Implications for Research and Practice

As more students with disabilities enroll in postsecondary education programs, higher education institutions must recognize and address the needs of their diverse populations, including those with ASD. For example, providing training to the faculty about universal design for learning to address the broad needs of all students. Letters from the Office of Disability Services to the faculty may provide more descriptions of the student needs and a list of useful resources. The college preparation of students with ASD must start during secondary education as part of their transition services. However, students with ASD and their parents report little student engagement in transition planning before college [52]. It is possible that an increased level of student engagement in the transition-services process can have them be better prepared to tackle the complexity of the new culture they have to navigate, especially during their first year. Higher education institutions should always find an alternative to reach out to their incoming students to share the available resources. The findings of this study add to the existing literature pinpointing differences within the population of college students with disabilities. Given the particular characteristic of a student with ASD, outreach programs should be developed to help them identify and use available resources, which should go beyond those offered through the disability centers.

## 7. Conclusions

This study provided preliminary results that depicted the unique challenges faced by students with ASD, such as low levels of risk-taking or social aspects of self-confidence; however, the results also showed that students with ASD report higher rates of the intellectual aspects of self-confidence. The findings revealed that when providing support and services for students with disabilities, it is essential to examine students’ strengths and support they really need, due to the varied characteristics of this population. Future research may further explore characteristics of different disability categories and establish the resources that are needed for students with disabilities.

## Figures and Tables

**Table 1 ijerph-18-11822-t001:** Individual, Familial, & School Characteristics of First-year College Students with Autism Spectrum Disorders and Learning Disabilities.

Variables	Autism Spectrum Disorder % (*n*)	Learning Disability % (*n*)
**Gender**		
Male	64 (23)	38 (113)
Female	26 (23)	58 (113)
Other	8 (23)	4 (113)
**Race/ethnicity**		
White	61 (23)	84 (118)
Asian	17 (23)	3 (118)
Hispanic	9 (23)	2 (118)
Black	9 (23)	2 (118)
Other	4 (23)	9 (118)
**Highest level of expected education**		
BA	18 (22)	29 (114)
MA	50 (22)	42 (114)
**Type of institution currently attending**		
2-year	26 (23)	22 (118)
4-year	74 (23)	78 (118)
**Full time**	91 (23)	99 (118)
**First generation college**	17 (23)	6 (115)
**Parents’ education level**		
Parent 1 College or above	74 (23)	83 (117)
Parent 2 College or above	74 (23)	77 (115)
**Parents’ income level (<$124,999)**	56 (23)	24 (108)
**Parent 1/Parent2’s employed**	87/83 (23)	83/63 (114)

Note: Total N is reported for each variable.

**Table 2 ijerph-18-11822-t002:** Self-rating of students with learning disabilities and autism spectrum disorders at the end of their first-year in college (numbers).

Variables	Learning Disability		Autism Spectrum Disorders		*p*-Value
	Average & Below Average	Above Average	Average & Below Average	Above Average	
Risk-taking	41	27	14	3	0.04
Self-confidence (intellectual)	34	35	4	13	0.72
Self-confidence (social)	36	48	12	5	0.1
Understanding of others	23	46	8	9	0.13

Note: The *p* value is from Fisher exact tests, there is no other test statistics.

**Table 3 ijerph-18-11822-t003:** Number of students who reported receiving remedial and academic support during their first-year in college.

Variables	Learning Disability		Autism Spectrum Disorders		*p*-Value
	No	Yes	No	Yes	
Remedial course	53	8	13	1	0.53
Academic support program	48	51	14	5	0.09

Note: The *p* value is from Fisher exact tests, there are no other test statistics.

**Table 4 ijerph-18-11822-t004:** Number of students with learning disabilities and autism spectrum disorders and their social and academic Interactions during their first-year of college.

Variables	Learning Disability			Autism Spectrum Disorders			*p*-Value
	Never & 1–2/Term	1–2 Month	Weekly & Daily	Never & 1–2/Term	1–2 Month	Weekly & Daily	
Academic advisor	57	23	30	11	6	6	0.79
Close friends at the institution	4	5	103	1	1	21	0.54
Siblings	9	17	45	3	3	12	N/A
Close friends not at this institution	22	13	75	6	3	14	0.52
Faculty during office hours	36	35	41	10	8	5	0.57
Faculty NOT during office hours	53	25	32	13	4	6	0.79
Graduate students	46	11	14	9	2	7	0.54
Parents	10	11	89	2	1	20	N/A

Note: The *p* value is from Fisher exact tests, there are no other test statistics.

**Table 5 ijerph-18-11822-t005:** Number of students with learning disabilities and autism spectrum disorders receiving support services first-year of college.

Variables	Learning Disability			Autism Spectrum Disorders			*p*-Value
	Not at All	Occasionally	Frequently	Not at All	Occasionally	Frequently	
Discuss course outside class	5	45	48	1	11	6	1
Tutoring	40	41	18	8	9	1	1
Group project	4	55	40	3	9	6	1
Study skills	53	30	15	11	4	3	1
Financial aid	67	24	8	13	4	1	1
Health services	61	33	15	10	7	1	0.01
Psychological services	62	33	5	13	3	2	0.56
Writing center	34	51	14	13	3	2	0
Disability center	39	29	31	10	2	6	0.24
Career services	53	42	4	11	7	0	1
Academic advising	21	60	18	7	9	2	0.48

Note: The *p* value is from Fisher exact tests, there are no other test statistics.

## Data Availability

Data is available by request at https://heri.ucla.edu (accessed 8 November 2021).

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
