# Peer review of "Students with Autism Spectrum Disorders and Their First-Year College Experiences"

_ijerph, 2021, doi:10.3390/ijerph182211822_

Round 1
Reviewer 1 Report
My main concern about this paper is the very low representativeness of the sample, that goes even beyond the low number of students with ASD included.
This small sample is actually a sub-sample of ASD subjects who decided to disclose their disability, who are in turn a small sub-sample of ASD subjects enrolled in the universities participating in the survey. This makes the small sample of 17 subjects a very far proxy of ASD populations enrolled in universities, raising the chance that this small sample is selected based on some unknown factors, biasing the results.
Besides this important weakness, I also think that several improvements are needed. I am here raising some general issues, but authors can find further comments in the attached document.
introduction: I found it too broad. I suggest to limit this part in order to provide the reader with the background needed to understand the aims of the paper but without making him to loose track.
methods: it is not clear how and when the two surveys are delivered to students, nor how the authors used data. It seems like the study has a cross-sectional design, but authors refer to the YFCY as a longitudinal instrument. Please clarify.
Figures: captions are missing. Where data in the figures come from? Do figures add something beyond what is described in the Results?
Table: please include both numbers and percentages. Also percentages seem wrong.
results: this part must be concise and better organized. No comments are to be included in this section.
Author Response
Thank you for reviewing our manuscript. Your comments and feedback helped us considerably improve the quality of our work.
Below is a list of the comments you have provided and how we addressed each comment in the revised manuscript.
My main concern about this paper is the very low representativeness of the sample, that goes even beyond the low number of students with ASD included.
This small sample is actually a sub-sample of ASD subjects who decided to disclose their disability, who are in turn a small sub-sample of ASD subjects enrolled in the universities participating in the survey. This makes the small sample of 17 subjects a very far proxy of ASD populations enrolled in universities, raising the chance that this small sample is selected based on some unknown factors, biasing the results.
- We address this comment in the limitations.
Besides this important weakness, I also think that several improvements are needed. I am here raising some general issues, but authors can find further comments in the attached document.
introduction: I found it too broad. I suggest to limit this part in order to provide the reader with the background needed to understand the aims of the paper but without making him to loose track.
- We have made changes to narrow the discussion
methods: it is not clear how and when the two surveys are delivered to students, nor how the authors used data. It seems like the study has a cross-sectional design, but authors refer to the YFCY as a longitudinal instrument. Please clarify.
- We have clarified in the introduction when the surveys were administered.
Figures: captions are missing. Where data in the figures come from? Do figures add something beyond what is described in the Results?
- We added the captions for the figure.
- We added additional information in the methods to clarify where the data is coming from
- Yes, the results present in details only the information about students with ASD but not for the LD as it is presented the figures
Table: please include both numbers and percentages. Also percentages seem wrong.
- Corrected %
results: this part must be concise and better organized. No comments are to be included in this section.
- We feel the results are concise and it is common to organize results by research questions. However, we understand that using research questions per se as headings for results is not common. So, we changed the research questions to statements as headings to better organize the results based on the research questions.
Reviewer 2 Report
- Subheaders in the lit review would be helpful for ease of reading
- 39-Wrong tense “Youth with ASD will graduate...in 2014-2015”
- 50-You say “decades of research” but citations only go back to 2006. That is not even two decades.
- 85-Dropping out, not dropping off
- 91-More description of ASD would be helpful here. This assumes some prior knowledge.
- 108-Be consistent with terms. You use autism and ASD.
- 167-More about the first-year experience for all students is needed. Before you can say that things are different for those with ASD, you have set the standard.
- You need something about learning disabilities in the lit review. You don’t even mention LD until your research question. Why are you comparing these two groups and not other disabilities? What is your rationale for that?
- 246-Specify that the characteristics you are referring to that are similar are demographic characteristics, not disability characteristics. Not clear.
- 272-A remedial program for what? There is a difference between a remedial program and a support program.
- 310-11-Statement needs a citation.
- 323-How do the differences in gender and ethnicity between the two disability groups at the college level compare to the bigger picture of those differences in K12 education or society at large? Why is it “worthy of further exploration”?
- 326 and 326-White not Whites
- 334-341-Whole paragraph does not really make sense.
- 342-”Given the characteristics of ASD…” I think you are using the term “characteristics” to mean both demographic characteristics AND disability characteristics. This is confusing. So when you say “given the characteristics of ASD” I don’t know if you mean demographic characteristics (like in line 246) or disability characteristics.
- 350-Wei, not we. This was confusing.
- 400-Need stronger/more concrete suggestions for practice based on your findings.
- Figure 1-% above average is confusing. Why not report actual rating scores?
- Figure 2-No explanation of the numbers on the y-axis.
- Research question #1-Family characteristics? Not clearly explained. I see in the table you have parent income and education but that is limited. What do you mean by family characteristics?
- ***The need to compare ASD and LD is not clear. You could have written this entire paper just with the ASD info and without the LD comparison and still provide suggestions for supporting students with ASD. How are suggestions for support for students with ASD different from students with LD? The rationale behind your decision to compare these two groups is not presented.***
Author Response
Thank you for reviewing our manuscript. Your comments and feedback helped us considerably improve the quality of our work.
Below is a list of the comments you have provided and how we addressed each comment in the revised manuscript.
- Subheaders in the lit review would be helpful for ease of reading
- We have organized the introduction to keep it concise; we appreciate the recommendation of using sub-headers, but given the changes we have made, we consider that sub-headers are no longer necessary
- 39-Wrong tense “Youth with ASD will graduate...in 2014-2015”
- I think it is the correct tense there.
- 50-You say “decades of research” but citations only go back to 2006. That is not even two decades.
- The word “decades” was deleted to reflect the citations
- 85-Dropping out, not dropping off
- We changed “dropping off” with dropping off”
- 91-More description of ASD would be helpful here. This assumes some prior knowledge.
- Added in the introduction
- 108-Be consistent with terms. You use autism and ASD.
- We are using ASD across the paper; changes were made to address the comment
- 167-More about the first-year experience for all students is needed. Before you can say that things are different for those with ASD, you have set the standard.
- Added in the introduction
- You need something about learning disabilities in the lit review. You don’t even mention LD until your research question. Why are you comparing these two groups and not other disabilities? What is your rationale for that?
- Added in the introduction
- 246-Specify that the characteristics you are referring to that are similar are demographic characteristics, not disability characteristics. Not clear.
- We have clarified that we are discussing the demographic characteristics.
- 272-A remedial program for what? There is a difference between a remedial program and a support program.
- We corrected “remedial program” with “remedial course”
- 310-11-Statement needs a citation.
- We added citation #35 to indicate that graduation for these students continue to be relatively low compared to peers without disabilities.
- 323-How do the differences in gender and ethnicity between the two disability groups at the college level compare to the bigger picture of those differences in K12 education or society at large? Why is it “worthy of further exploration”?
- Although this question is interesting; it is however beyond the scope of this paper as it focuses on postsecondary education. Adding this information is an overstretch of its focus.
- 326 and 326-White not Whites
- Corrected
- 334-341-Whole paragraph does not really make sense.
- We rephrased the sentence to make it clearer.
- 342-”Given the characteristics of ASD…” I think you are using the term “characteristics” to mean both demographic characteristics AND disability characteristics. This is confusing. So when you say “given the characteristics of ASD” I don’t know if you mean demographic characteristics (like in line 246) or disability characteristics.
- 350-Wei, not we. This was confusing.
- Corrected
- 400-Need stronger/more concrete suggestions for practice based on your findings.
- We added a couple of sentences to provide examples about training faculty on universal design for learning and enrich the letter to faculty from the office of disability services.
- Figure 1-% above average is confusing. Why not report actual rating scores?
- Corrected in Figure 1 and Figure 2
- Figure 2-No explanation of the numbers on the y-axis.
- Corrected
- Research question #1-Family characteristics? Not clearly explained. I see in the table you have parent income and education but that is limited. What do you mean by family characteristics?
- Corrected
- ***The need to compare ASD and LD is not clear. You could have written this entire paper just with the ASD info and without the LD comparison and still provide suggestions for supporting students with ASD. How are suggestions for support for students with ASD different from students with LD? The rationale behind your decision to compare these two groups is not presented.***
Addressed in the intro, before the research questions
Round 2
Reviewer 2 Report
Line 41-42 "Will have graduated from 2014-15."
96-Hypersensitivity, not hypersensibility
162-Still need citations
169-Still no clear description of LD
213-Need explanation of crosstabs as a data analysis method. That whole paragraph is one long sentence.
Figure 1-These self-ratings are a % of what? % of students who report being a risk-taker? Your figure descriptions are not clear. Also, your below-average, average, and above-average notations are not clear.
Table 1-What is your N? You have %ages of male and female, etc, but not what they are percentages of. How many participants did you have? If I recall correctly, you had this in your first draft so why did you remove it? If it was because your ASD N was low, changing it to %ages will not change that flaw.
264-Again, what is remedial? You do not explain what a remedial course is and how it is different from an academic support program (260_
Figure 2-There must be an easier way to report the data than this car graph. It is too hard to interpret. Same with Figure 3.
319 and 325-Do you mean demographic characteristics?
420-This is not a conclusion This is a repeat of your abstract.
Author Response
Line 41-42 "Will have graduated from 2014-15."
- We have made the corrections in text
96-Hypersensitivity, not hypersensibility
- We have made the corrections in text
162-Still need citations
- Citations were added
169-Still no clear description of LD
- More sentences were added to explain why we have chosen LD as a comparison category.
213-Need explanation of crosstabs as a data analysis method. That whole paragraph is one long sentence.
- We have included additional information on crosstabs.
Figure 1-These self-ratings are a % of what? % of students who report being a risk-taker? Your figure descriptions are not clear. Also, your below-average, average, and above-average notations are not clear.
- As per reviewer 1 suggestion, we have changed all Figures to tables and included the N instead of %. We better defined the tables.
Table 1-What is your N? You have %ages of male and female, etc, but not what they are percentages of. How many participants did you have? If I recall correctly, you had this in your first draft so why did you remove it? If it was because your ASD N was low, changing it to %ages will not change that flaw.
- Changes were made to table 1 by adding both % and total number for each variable.
264-Again, what is remedial? You do not explain what a remedial course is and how it is different from an academic support program(260_
- HERI did not provide definitions for these 2 concepts; a link to all the instruments used to collect data we have used in this study is provided at page 5, line 225.
Figure 2-There must be an easier way to report the data than this car graph. It is too hard to interpret. Same with Figure 3.
- As per reviewer 1 suggestion, we have changed all Figures to tables and included the N instead of %. We better defined the tables.
319 and 325-Do you mean demographic characteristics?
- We made the adjustments in text.
420-This is not a conclusion This is a repeat of your abstract.
- We included a new paragraph for conclusion.
Note: we have adjusted the numbers each reference in the reference list
Round 3
Reviewer 2 Report
This is much improved. I would still like to see a stronger conclusion with specific recommendations.